# The Role of Shared Resilience in Building Employment Pathways with People with a Disability

Perri Campbell [1,*], Erin Wilson [1], Luke John Howie [2], Andrew Joyce [1], Jenny Crosbie [1] and Robyn Eversole [3]

1   Centre for Social Impact, School of Business, Law and Entrepreneurship, Swinburne University of Technology, Hawthorn 3122, Australia; ewilson@swin.edu.au (E.W.); ajoyce@swin.edu.au (A.J.); jcrosbie@swin.edu.au (J.C.)
2   School of Education, Deakin University, Geelong 3216, Australia; luke.howie@deakin.edu.au
3   Freeman College of Management, Bucknell University, Lewisburg, PA 17837, USA; r.eversole@bucknell.edu
*   Correspondence: pcampbell@swin.edu.au

**Abstract:** For workers living with a disability, pathways to sustainable employment in the open labour market are inhibited by barriers operating at different structural and societal levels. The culture of Australia's government employment services has applied a 'work-first' approach that emphasises finding people employment rather than supporting the acquisition of skills and education. The net effect of this approach is the preferencing of short-term employment solutions, with a focus on individual behaviour or so-called resilience and an emphasis on personal responsibility instead of addressing structural issues. In this paper, we explore how people with disability can be supported in finding employment through a shared resilience approach offered by a Work Integration Social Enterprise (WISE). We suggest that WISEs can provide the conditions for shared resilience by developing and sustaining networks needed to generate hybrid pathways to work and by role modelling inclusive work conditions in the open labour market.

**Keywords:** work integration social enterprise (WISE); Australian disability enterprise (ADE); open employment; supported employment; customised; disability; hybrid; tailored employment; resilience; neoliberalism

## 1. Introduction

For workers living with a disability, pathways to sustainable employment in the open labour market are inhibited by a range of barriers operating at different structural and societal levels [1–5]. In Australia, where 4.4 million people live with a disability, representing just under 18% of the population, the focus is often on the issues experienced by individuals living with disability and not the systemic conditions that people with disabilities face [6]. Some people with a disability in Australia work in 'sheltered workshop' conditions, which separate and segregate individuals from the broader working community [7] (p. 227). In Australia, sheltered workshops have been referred to as Australian Disability Enterprises (ADEs), which offer employment specifically for people with a disability. Recently, some ADEs have begun to identify themselves as social enterprises or Work Integration Social Enterprises (WISEs). ADEs offer supported in-house employment and training for people with a disability. While WISEs do this, they also offer employment pathways to the mainstream labour market for people with experiences of marginalisation, including but not limited to people with a disability.

The culture of Australia's government employment services has traditionally applied a 'work-first' approach that emphasises finding people employment rather than supporting the acquisition of skills and education. The net effect of this approach is the preferencing of short-term employment solutions, with a focus on individual behaviour or so-called resilience instead of addressing structural issues [8,9]. Resilience here refers to the capacity of individuals to bounce back in the face of adversity and respond to challenges and

opportunities through innovation and entrepreneurialism [10] (p. 185). This enables a reduced role for government services and an emphasis on personal responsibility (i.e., individuals are 'responsibilised') [6] (p. 17)] [11]. In this paper, we explore how people with disability can be supported in finding employment through a shared resilience approach offered by one case study Work Integration Social Enterprise. Shared resilience recognises that individuals should not bear the full weight of entry into the labour market. Instead of expecting individuals to be responsible for solving broader social problems, such as the historical exclusion of many people with a disability from the labour market, the concept of shared resilience suggests that organisations, such as employers and WISEs, play a role in creating the conditions for resilience. Shared resilience is the dynamic process through which resilience is created by individuals and organisations working together to overcome disadvantages. This paper is based on a two-year research project conducted with a WISE located in regional Victoria in Australia's southeast that offers supported employment in-house and pathways to open employment to people with a disability. The research questions guiding our work were (a) what are the barriers people with a disability encounter in transitioning from supported to open employment, and (b) how does a large WISE operating in the disability sector support individual pathways from supported to open employment? In what follows, we explore the literature that frames the concept of an ideal worker and how it relates to the historical marginalisation of people with a disability in the open labour market. We then discuss how the Australian disability support sector and wage assessment have shaped and limited employment options for people with a disability. We suggest that WISE approaches can potentially play a role in generating supportive pathways to open employment. Our case study organisation used a shared resilience approach to build a network with each individual to help carve out an appropriate, supported, and flexible employment experience and pathway. We outline our methodology and methods used for gathering data with our case study organisation. In our data section, we show how employment pathways are fraught with structural and interpersonal challenges, as well as systemic contradictions that create barriers for people with disabilities from transitioning into the open labour market.

We suggest that models for disability employment embrace 'tailored follow-ups' as an essential step in the transition process [12] (pp. 3, 9–10). This generates the possibility of tailored returns and hybrid work options, which enable people to move between supported and open employment as their life circumstances change. Tailored returns are a structured pathway that allows an individual to move between employment options (i.e., back and forth between a WISE and open employment), retain their networks across these spaces, and build new networks in the community. We conclude by suggesting that such disability employment approaches challenge established ideas of the worker through a 'shared resilience' approach in which the WISE and employer work together to cultivate the conditions for worker resilience at the individual and organisational level.

### 1.1. Disability, Employment, and the Ideal Worker

Australian census data show that there are approximately 4.4 million people living with a disability, representing just under 18% of the population [13]. This is based on a definition of disability which reflects 'any limitation, restriction or impairment which restricts everyday activities and has lasted, or is likely to last, for at least six months' [14]. For people aged between 15 and 64, data before COVID-19 showed that people with a disability have lower rates of labour force participation compared to people without a disability (53.4% compared to 84.1%) [14] and higher unemployment rates (10% compared to 4.6%) [15]. Data also show that graduates with a disability take longer to achieve full-time employment than graduates without a disability [16]. Employment rates for people living with disabilities in Australia are stagnant. Australians remain well behind most other Organisation for Economic Co-operation and Development (OECD) countries in labour market participation for people with disabilities [17]. Employment outcomes for those groups most frequently employed in ADEs are even worse. 'In 2017, people

with intellectual disability comprised 70–75% of the ADE workforce (DSS, 2018), and in 2020, the National Disability Insurance Agency (NDIA) reported that National Disability Insurance Scheme (NDIS) participants with an intellectual disability over 25 years of age were predominantly employed in ADEs (70%) if employed (NDIA, 2020)' [18,19] (p. 4). Down Syndrome Australia reports that the majority of people with Down syndrome who are employed are working in Australian Disability Enterprises (ADEs).

The notion of the 'ideal worker' helps explain, in part, why this workplace inequality persists. Despite supposedly embracing contemporary (human resource) managerial notions of equity, diversity and inclusion, 'flexibility,' and 'independence', the post-twentieth-century workplace continues to reproduce and generate new forms of inequality [20] (p. 271). Feminist sociologists have critiqued the notion of the 'ideal worker'. For Acker [21] (p. 151), organisations reproduce cultural and social norms that seek to promote the idea of the 'abstract, bodiless worker' that 'occupies the abstract, gender-neutral job [that] has no sexuality, no emotion'. This situation likely emerges from the 'development of large, all-male organizations that are the primary locations of societal power' that, in practice, had little 'historical' imagination for including women, different sexualities, other races and languages, as well as anything other than a traditionally abled body. Or, as Acker [21] (p. 151) described it, the 'suppression' of difference 'in the interest of organization and the conceptual exclusion of the body as a concrete living whole'.

The body as a social formation provides the platform for understanding the intersection of workplace dynamics and how these are shaped by contemporary forms of governance, such as neoliberalism. 'Resilience' is the key neoliberal trait honoured in work and social environments alike. Resilience is about being able to bounce back in the face of challenge and uncertainty. Those who are resilient 'are imagined as being able to respond to these challenges and opportunities through their capacities for innovation and enterprise. . . and dispositions that enable them to thrive in contexts of uncertainty and disruption. They are resilient' [10] (p. 185).

Following the COVID-19 pandemic and the new understandings of mental illness in its wake, organisations seem to have a greater awareness or appreciation of the need for care, 'inclusion', and diversity in workplaces [22,23]. Yet the figure of the ideal worker problematises contemporary inclusion goals as many organisations still seek employees who do not disrupt but maintain organisational social order, which is historically able-bodied [4].

Research into low employment rates for people living with a disability has long emphasised performance and ability as key criteria in decision-making around hiring. Soldatic [24] (p. 51) explores the impact of neoliberalism on the bodies and employment opportunities of people with a disability. Bodily mobility under a 'neoliberal workfare state' becomes a way of classifying the capability of people with a disability. Soldatic [24] (p. 1) defines disability as a 'socially constitutive collective class and identity with the emergence of the postmodern, neoliberal nation state'. Mobility, in this context, is operationalised through time-sensitive capacity tests tied to new forms of labour market participation.

> *Under the temporal practices of the 2005 welfare to work legislation, two new classes of disability emerged: those 'partially mobile' disabled subjects who signified momentary immobility through a current inability to work (but had the bodily temporal dispositions to actively work towards a state of improvement) and those 'fixed' disabled welfare subjects who remained permanently excluded from the improvement of the nation because of their 'continuing inability to work'.* [24] (p. 74)

The marginalisation of disabled people from the open labour market is perpetuated in Australia via a classification of bodies and citizens, which draws on medical technologies and time-based work tests. For Soldatic [24] (pp. 76–79), the temporal work test is a problematic tool that requires critique and reinvention:

*The temporal relationship between disability social entitlements, state classification regimes and practices of social exclusion is likely to fluctuate with each temporal murmur or seismic shift.*

*The resultant temporal socio- classification process is underpinned by relations of exchange, and it is through this process that disabled people and their bodies become inscribed with value.*

Positioning mobility and capacity against existing time-sensitive benchmarks of the 'able-bodied' or ideal worker has the effect of centring and reaffirming ablest benchmarks.

Foster et al. [25] (p. 705) argue that 'standard jobs, designed around this 'ideal' creates a mismatch between a formal job description and someone with an impairment' and that this mismatch leads to resistance from organisations to implementing adjustments and including employees with disability in the workplace. Østerud [4] (p. 1) argues, however, that social factors are of 'stronger emphasis' when hiring decisions are made between an applicant living with a disability and an applicant without: 'Social cohesion concerns can lead to disabled people being rejected based on these prejudiced impressions and their lack of 'fit' with the organisational culture' [4] (p. 4). When ideals of 'cohesion' are prevalent in hiring processes, more 'socially competent' people are favoured [4] (p. 4, and see [26]). This may prove especially challenging for people who require support both inside and outside of workplaces where social values around being able to live independently are considered a vital marker of socio-cultural belonging.

Barnes and Mercer [1] argued that disability employment should be understood in the context of independent living. Encoded within discourses of employment opportunities for people living with disabilities is the idea (or ideal) that such employment should be independently performed, with minimal supports at some point in the employment journey. This becomes especially important in the context of unsupervised employment, where work allocations are met within timeframes achieved with minimal direction or guidance from managers and leaders [27]. The outcome of these conditions is employers hire a certain type of worker—one that involves an 'abstracted image' of what an ideal worker should be—and, in so doing, unconsciously 'marginalising minorities who the employers imagine will have a more difficult time naturally fitting into the group and existing social practices' [4] (p. 4).

### 1.2. Employment Pathways for People with a Disability: ADEs, Supported and Open Employment

In response to this problem of the 'ideal worker' and consequential marginalisation of many people with a disability, Australian legislation and policy have tended to bifurcate employment into open (i.e., emphasising independence) and 'supported' employment, emphasising high support needs [28] (p. 2). Employment options are often framed as either employment in an Australian Disability Enterprise (ADE) or facing the myriad of assumptions about what constitutes ideal workers in the open labour market. These responses have had the consequence of segregating people living with a disability from others, creating a dichotomy of normalised and abnormalised bodies.

Historically, ADEs offered employment for people with a disability at a time when community-facing or open employment opportunities were not available. ADEs provided ongoing supported employment. However, the perpetuation of the historical ADE structure has contributed to the conditions for segregated work settings and halted transitions into mainstream or open employment [28] (p. 1). Many ADEs do not focus on generating an employment pathway out of the ADE into the open labour market because this is not an inherent part of their organisational structure. Recent policy and funding shifts attempt to alter this employment landscape. Funding for ADEs has changed since the creation of the National Disability Insurance Scheme (NDIS), which shifted from 'case based' funding directly provided to ADEs to individualised funding [28] (p. 3). This means that individuals can purchase support to work in any workplace, not just ADEs. However, the market of employment service providers (i.e., Disability Employment Services, DES) to utilise

this funding for employment pathways is yet to emerge. Additionally, employers require coaching and confidence building to break established employment patterns and 'ideals'.

As noted in the previous section, Soldatic [24] argues that one of the key challenges is the temporal assessment of disabled bodies for work. Temporal assessments include wage assessments, which ultimately restrict access to employment. 'Criteria that restricts entry to employment supports based on number of hours worked per week is likely to mean that those most in need of employment support are routinely ineligible for it' [28] (p. 5). Wilson et al. [28] (p. 7) propose a remodelling of wage assessment via a biopsychosocial approach. Rather than attempting to locate 'capacity' in the 'impairments of the individual', this alternative approach would re-focus assessment on 'identifying the wide diversity of factors restricting work participation' [21] (p. 21).

> *The biopsychosocial approach fundamentally reshapes the way 'work capacity' is understood and requires a wide range of employment supports to be offered to all people with disability to best address the barriers to work experienced by the individual.* [21] (p. 7)

Wilson et al. [28] call for no less than a re-alignment of supports with the lived experience, work, and social environment that individuals occupy. This conceptual framework shifts the focus from the individual and their capacity to include the employment landscape itself. This reconfiguration raises questions about who is responsible for creating the conditions for work capacity and helps to build the foundation for our understanding of shared resilience in the section that follows.

### 1.3. Work Integration Social Enterprises and Shared Resilience

Among the approaches adopted by Australian employment service organisations to tackle broader employment challenges is the Work Integration Social Enterprise, or WISE, model. Davister, Defourny, and Gregoire [12] (p. 3) describe WISEs as 'autonomous economic entities' that seek 'professional integration' for people who experience 'serious difficulties' in the labour market. With a combination of 'productive activity' and training and 'tailored follow-up', WISEs seek to enhance the prospects for people who wish to hold employment in the open labour market [12] (p. 3).

Research evidence shows that social, economic, and health benefits flow from participating in social enterprises [29,30]. When they work well, these workplaces can open pathways into employment and education and build confidence, social connections, and feelings of self-worth. These outcomes improve health and well-being. Securing open employment (the goal for most WISE-based programs) is associated with a range of positive outcomes [31]. Smith et al. [32] (p. 59) argue that 'though a challenging undertaking, Social Enterprise provides a promising employment option for some people with ID [Intellectual Disability], when such initiatives are driven from executive and senior personnel of an organisation'.

In recent years, ADEs in Australia have begun to identify or re-structure as Work Integration Social Enterprises (WISE). One distinct difference between WISEs and ADEs is that WISEs orient their activities towards employment pathways out into the open labour market. The degree to which ADEs align with the WISE model varies from organisation to organisation. Those ADEs building pathways to the open labour market are more in line with the WISE model. These WISE essentially provide people living with disabilities an opportunity to undertake paid work in a supported environment to gain skills, experience, and confidence before attempting employment in the open labour market [12,33]. The ADE service is useful, particularly for those who 'need substantial ongoing support to obtain or retain paid employment' and where competitive employment at award wage is 'unlikely' [13] (Part II, Div 1, 7, see also [34]). WISE, when developed with these goals in mind, blends the benefits of 'supported' and open employment. Emerging in this space is a third option for employees with a disability: employment within the operations of the WISE itself [31,35] (pp. 225–226).

There are several mechanisms available to provide supports in open employment settings that involve a broader understanding of the employment landscape; however, they

have not been widely utilised by ADEs to generate open employment pathways [28] (p. 1). In particular, Davister et al. [12] (pp. 3, 9–10) and Defourny et al. [33] have explored the need for a 'tailored follow-up' for successful WISE programs, which involves staying in touch with individuals to actively follow their employment journey. Key mechanisms that include a follow-up step are customised employment, job carving, and tailored employment [36]. They involve brokering relationships with employers and developing their knowledge of appropriate supports, understanding how an individual works best, and understanding how capacity can be matched with market demands and business needs. Customised employment involves, first, creating or identifying a job, or parts of a job, that the individual is interested in doing and, second, putting in place strategies for mastering and using the equipment, tools, or work setting that the job requires. This process is sometimes confused with job carving, repeated tasks, and standardised sectioned-off tasks. However, customised employment is a particular approach to employment guided by a structure and fidelity scale and has the potential to improve productivity and output [37]. Carving and tailoring a job involves sectioning off specific parts of one role or adjusting the role to best match the individual. A tailored approach addresses the interests and needs of both the employee and employer [38] (p. 21).

Individuals seeking open employment face a range of challenges, including temporal wage assessment and funding issues, an ill-equipped labour market, limited employer knowledge and confidence, and limited service providers. The supported employment mechanisms we describe above can help to address some of these challenges by taking the *employment landscape* into consideration alongside the individual. This kind of biopsychosocial approach acknowledges the structural and systemic nature of marginalisation from the labour market and encourages us to imagine work capacity beyond the body of the individual. Work capacity is also attached to the role of employers, service providers, and WISEs. By challenging expectations of the role of workers and employers alike in developing employment pathways, the figure of the 'ideal' worker is unsettled. This is where we see the grounds for shared resilience emerge through the problematisation of existing neoliberal narratives of resilience and entrepreneurialism [38] (p. 4). The biopsychosocial approach suggests that work capacity is not solely the remit of the individual as resilience narratives would have us believe. As we will discuss in the sections that follow, WISEs are able to share in the responsibility of employment pathways, and this type of intervention is evident in our research in Australia.

## 2. Methodology

In 2021, we received funding (from the Department of Social Services) to adapt a WISE organisational design model for a disability employment organisation to support the employment pathways of people living with a disability. This WISE model defines organisational categories such as 'organisational structure', 'culture', and 'relationships' that are important to the successful functioning of WISEs. The organisation we partnered with provides supported employment via a number of WISE settings for people with a disability in industries such as hospitality, warehousing, and landscaping, and they also offer pathways to open employment.

We carried out action learning research with the participating organisation from November 2021 to March 2022. Action learning involves learning from experience through a process of observation, reflection, planning, and acting. It allows knowledge and information to be shared between different individuals and groups as part of a 'process of change' without things having to be 'fully worked out in advance' [39] (p. 256). As such, it is a suitable method for capturing and utilising dynamic organisational changes and developments as vehicles for learning through reflection [40,41].

The research was collaborative, involving organisational managers and supported and non-supported employees at this large disability employment organisation in regional Victoria, Australia. In total, 27 interviews were conducted alongside four action learning workshops (involving supported employees, non-supported employees, and managers)

and five steering committee meetings (involving stakeholders from businesses that have employed people with disabilities in open employment, prospective employers, disability industry experts, people with disabilities who have worked in open employment, and family members/carers). The majority of interviews were conducted with staff and supported employees who were not part of either the action learning group or the steering committee.

A selective and iterative coding approach was undertaken to identify key themes from interviews regarding employment barriers, facilitators, and how the organisation provided employment and support to people with a disability. The initial coding framework was structured by the WISE model elements and sub-elements (available here: https://socialenterprisewellbeing.com.au/index.php/insights-2/, accessed on 10 November 2023). The overall effect is the provision of wraparound, networked support for tailored, hybrid employment pathways.

In this paper, we focus on the data that speak directly to the concepts of shared resilience and tailored returns. These concepts were identified against the organisational elements (structure, culture, funding and finance, and pathways) and coded against child nodes (or sub-themes): 'culture, stigma and discrimination', 'employer disability know-how and awareness', 'support and lack of support', and 'funding and finance'. These child nodes form the themes that are unpacked in our data and analysis section. Our analysis of themes involved 'reviewing the coded data extracts for each theme to consider whether they appear to form a coherent pattern' and whether 'the themes accurately reflect the meanings evident in the data set as a whole' [42] (p. 9).

In the sections that follow, we draw on the literature explored in this paper and our conceptual framing of shared resilience to critically engage with responses from supported employees, disability support staff, managers and trainers, and employers and stakeholders in the open labour market. The first data section explores the employment experiences and challenges of people with a disability, and the second data section explores how these can be managed and reimagined via a shared resilience approach.

## 3. Findings

### 3.1. 'Don't Worry, Mate, Get on with Your Job': Social Stigma, Employer Confidence, and Open Employment Funding Challenges

*I don't want to stay here the rest of my life. I want to go out there in employment and socialise and talk to people. (Patrice, November 2021)*

The organisation where this research was conducted has experience supporting people living with a disability to work in-house at the WISE and to find work in open employment. The supported employees we spoke to identified key challenges in open employment, including stigma and discrimination, employer knowledge and confidence barriers, lack of support and preparation/training, and funding disincentives. In this section, we discuss these challenges to bring to light the spaces where a shared resilience approach can be beneficial.

Experiences of stigma and discrimination are all too common in open employment. Max is a young man with an intellectual disability working at the WISE. We asked him what it is like to work in open employment and if there are any challenges. Max describes an experience in open employment shaped by a number of pressure points, including poor customer attitudes, employer confidence and knowledge, and training and preparation for employment. Max says that customers, often in a hurry, would ask him a question, and if he was unable to immediately answer, customers would just walk away or get angry or even swear at him. He had discussed some of these issues with his manager in the past but felt unsure how to handle these interactions with customers and, in this case, felt ill-equipped to approach a manager to discuss the issues further; he says:

*In open employment I feel like I'd be on my own a little bit with a situation like that. Like, I could talk to my boss, but most bosses would say, "Don't worry, mate, get on with your job", or something like that. I think he would say, "It's your own issue, deal with it in your own time", or something like that. (Max, November 2021)*

Max's concerns about customer attitudes, limited support, and understanding are commonly experienced. For many in the disability sector, this lack of support is grounds for concern. Disability support worker Jean (support worker, November 2021) believes that prospective employers must make reasonable accommodations when employing people who live with disabilities. The benefits for organisations in developing an inclusive workplace culture can be significant:

> *For any employer that wants to employ a person with a disability, they've got to be on board in regards to a cultural perspective. Absolutely has to be critical that they want to make a difference and be a positive influence on this person's life and have empathy. (Jean, support worker, November 2021)*

This inclusivity must filter from the top to every facet of an organization. Trainer Grace says:

> *Success really does rely on the other staff and the culture of the organisation or company and that's why sometimes those smaller companies are really good, because it might be an owner operator, which is less likely to sort of turn over [staff]. Or if it's a bigger company— it really needs to be embedded into the culture. (Grace, training staff, October 2021)*

Patrice, a supported employee with an intellectual disability, describes the intersection of inadequate planning and customising of the employment role, a lack of appropriate supports, and social stigma. This was her experience before joining the WISE and receiving support to carve out a pathway to open employment:

> *I think the problem is when people with disability go out into open employment, I think they're scared. Because when they go out there, people with disability, "oh, they can't do this, and they can't do that". And they get really nervous. But once they're here [supported work], they've got people to help them too. I've been watching, and since I've got my confidence up, and then when you go out in employment, they get all timid. I thought, "Oh, shivers, when I go in employment, where do I go?" I thought, "With a disability, what do I do?". (Patrice, November 2021)*

Lack of preparation is a key challenge. Kelly, a supported employee, describes feeling like she was 'thrown in the deep end'. She was working during a busy time of the year and says she was not offered training or support in her customer service role:

> *I thought I was selling the products but I was on cashier, which is not my strength, with money. And I was only working there for two weeks. They'd showed me one time how to use the till, but… I work by being shown things a couple of times. They walked in front of me and did it themselves, didn't explain what they were doing or anything… they didn't give me a chance, so I was only working there for two weeks…*

> *I wasn't really comfortable with that and I was always calling and asking, "How do I do this, how do I do that?". (Kelly, December 2021)*

The support available in the workplace is shaped by employers' own knowledge and confidence in employing someone with a disability. The challenge here is that many employers have limited experience in working with people with a disability and, as such, have limited opportunities to develop knowledge and confidence in this area. As one employer tells us:

> *I think… for us as a business… we have to acknowledge that our warehouse workers… are basic trained warehouse staff. They are not trained in dealing with any challenges [in employee behaviours]. (Jacqui, employer, January 2022)*

Organisations employing people with a disability should allow for the time needed to provide appropriate support and training in the new environment. There can be particular areas where people face challenges and require some additional support, including

> *Time management. Dealing with difficult other employees or customers. Being punctual. Being consistent in the work they do. Good hygiene. Having a work/life balance and not*

*relying on others to assist. Expectations. Obligations. Working to a deadline, or working to a budget. (Jean, support worker, October 2021)*

This level of support is often dependent on disability support provided via Disability Employment Services (DES) and individual National Disability Insurance Scheme (NDIS) funding [18]. As one disability services staff member says:

*. . .There is a lot of conversation about holistic supports and all of the extra things that they [NDIS] could be doing. . . but the way that [NDIS] are often modelled limits the amount of support they can provide each jobseeker. Our supported employees moving into open employment and their new employer often require intensive supports, and within the workplace, particularly at the start. This is often outside of the scope of what a DES [disability employment services] provider can assist with. (Grace, training provider, October 2021)*

For Grace, some support structures and funding should be devoted towards supporting a business or organisation to employ, support, and retain people living with a disability in open employment. Yet, the opposite is often true.

The challenges our participants described range from dealing with social discrimination to a lack of employer knowledge, support, and inadequate preparation for employment roles. Entering open employment renders someone living with a disability more capable and more able in the minds of people employed by regulatory authorities who manage disability supports and disability pensions (DSP). People who are able to attempt open employment are eligible to receive some supports; however, this support is often not intensive enough for moderate- and high-need individuals, and it may be reduced over time. The concern is that individuals requiring ongoing support enter open employment and have their financial and employment supports (like DSP) significantly decreased. This limits individual capacity and choices. The danger is that individuals are left with reduced funding and unable to maintain participation in a WISE or open employment.

*3.2. 'It's a Step-by-Step Process': Shared Resilience, Support, and Customisation in Employment*

The WISE was able to provide appropriate workplace conditions and supports for people with a disability in-house at their WISEs and in open employment settings. The challenge, in many respects, is translating those aspects of this organisational setting and approach in the open labour market. Uniquely, the organisation operates an outward-facing employment services division that focuses on securing open employment opportunities for people with a disability. Many disability-focused WISEs in Australia do not have this organisational advantage. While the employment services division undertakes much of the external-facing communication and set-up work by coaching employers and brokering tailored positions, creating the right conditions for work also relies on the labour of the individual person with a disability and their individual support worker/coordinator. In order to generate a sense of 'shared resilience', responsibility and accountability must be held (or shared) by the individual and a number of actors in the individual's life. As new actors (open employers) are embedded in an individual's network and employment journey, there is potential for a stronger intervention into the conditions that render open employment individualistic and competitive. In what follows, this sense of shared resilience is described by our interviewees, supported by mechanisms such as customised employment, job matching, job carving, and tailored employment.

Interviewees reported that their best chance to thrive in open employment was if they worked with friendly and helpful colleagues and managers, were provided opportunities to engage in balanced teamwork, and received on-the-job assistance (i.e., from support workers) in an appropriately accessible work setting. Supportive employers were those who could provide accessibility resources and tools that help with understanding the role, incorporate flexibility (i.e., time allowances for breaks), and a "go-to" person or buddy who could be asked questions about the role. In this section, we explore these modes of support and the client/employee and employer relationship.

### 3.2.1. The Role of Customisation in the Employer–Employee Relationship

Maintaining open communication with caregivers and family members creates a strong network. Yet it is equally important to develop a local, place-based network with an individual in their community to create community-facing, local, open employment opportunities [42]. The relationships that individuals develop within their working community are vital to personal development, network building, and future employment options. Family and community relationships can create a foundation for appropriate employment conditions to emerge.

Employers can become a key component of a shared resilience network. Strong relationships are built over time and factor into the needs of both employer and employee to establish goals and knowledge and break down social stigma barriers. WISEs can develop these relationships by providing services and through partnerships with new and existing business clients, as one staff member tells us:

> *I reached out to them, and before you know it, we've got a job painting... it's just getting the conversation started, but where does it end? There is really no endpoint. (Grace, interviewed October 2021)*

Preparing employers by communicating the support requirements of individual employees is an important step in building a network. This may begin as an informal conversation with an employer but can also take the form of customising a role for an individual.

As employment and education stakeholders told us, identifying the interests and goals of individuals is key:

> *I think finding what they like to do and what interests them is key because if they're interested in something, they will know it very well. (Mark, employer, November 2021)*

> *I think it's a step-by-step process of having someone understand where they want to go, how they might do it, and the confidence to be able to take those steps without being burnt along the way. (Anthony, education stakeholder, October 2021)*

Customising employment involves understanding and identifying how individuals work best and what forms of support are required. For instance, many supported employees enjoy variety and require support and training to adapt to new tasks or jobs. Some employees prefer more lead in time before their work tasks or work location are altered, as one employer explained:

> *...in terms of when she got moved around, it really unsettled her. So, I think just to have that knowledge, if we tell our Warehouse lead, "Don't move this person around", she won't get moved around. (Jacqui, employer, January 2022)*

This kind of understanding can be developed with the ADE, as Dave, the trainer, explains:

> *If you've got an employer that's taking on one person and giving them a role within their organisation, they can actually be specific and very tailored around what that person needs. And so then, they just need to have the tools and the toolbox... and they've got to be willing to invest in that to set up the environment. (Dave, manager/trainer, October 2021)*

For WISE staff, an employer's 'toolbox' is shaped by an understanding of the individual employee:

> *They've got to understand around—there's got to be some training around that individual. So, I guess the ability, if you've got an employer that's taking on one person and giving them a role within their organisation, they can actually be specific and very tailored around what that person needs. (Dave, manager/trainer, October 2021)*

The other side of setting up a customised role involves understanding the employer and their needs [43]. Employment services staff planned and practised a targeted approach, which involved meeting employers to develop strategies for engaging employees for tasks and roles. As one staff member advises:

*. . .if we're going into an employer, we don't want to be ambiguous. . . So, if I was going to go into a factory, I would've already thought about what tasks would happen in this factory. . . you just be very specific about some ideas of things that our participants could do to generate those conversations in the beginning. (Lauren, employment support staff, December 2021)*

Trials and training for both the employer and the employee can help all stakeholders better understand how to customise the workspace and find the right fit between work tasks, individuals, and organisations:

*We'll do a trial day, whether it's two hours, three hours, four hours, we'll bring our supported employees to you. We'll trial the work, which means our guys can feel it, touch it, see if it meets our scope of work. (Alan, trainer/manager, October 2021)*

Customising and job carving must also be accompanied by meaningful and purposeful engagement with work goals [44]. Understanding, for instance, how smaller tasks contribute to the whole or other goals of the organisation/business can improve confidence, motivation, and well-being.

Customising work with individual interests and strengths boosts confidence, well-being, and skills, particularly when work is coupled with appropriate training [45]. Connor is an employee who transitioned into open employment after training with the WISE for a number of years. He had trained in land care and now experienced confidence and a sense of belonging in his open employment land care role. When asked if he was happy in the role, Connor replied, 'Yeah, loving it. . . I wish I had got it earlier' (Connor, open employment, October 2022).

Employers who show empathy, attempt to understand the individual they are supporting, and commit the time required to support workers with a disability are able to create strong relationships with both the individual and their support network and service providers. Many of the staff we spoke to felt that employers who had a hiring policy of diversity benefited from the contribution supported employees make to an organisation's performance and culture.

3.2.2. 'You're Not Looked at or Judged Here. . .': Creating a Culture of Respect

A respectful and understanding workplace culture plays a significant role in establishing and maintaining a sense of shared resilience. For instance, teamwork is a practice of shared resilience common to many work environments and relies on the allocation of appropriate roles to different individuals. A shared resilience culture emphasises innovative team roles and enhanced strengths-based approaches (i.e., customised and tailored). Underlying values that generate this kind of workplace culture include mutual respect, trust and privacy, understanding and flexibility, and high-quality standards for products and services [38] (p. 47).

Respectful workplace cultures acknowledge the rights of employees to privacy and take a strengths-based approach to training and coaching. The supported employees we interviewed reported feeling respected in their roles because they were 'not looked at or judged like you have a disability' (Michelle, supported employee, December 2021). This was largely because the WISE staff were able to ensure that employees were valued, their accomplishments acknowledged, and their voices heard in the workplace. Staff commented:

*Yes, we support them and everything, but we're becoming more like an Open Employment style business because we're getting more staff and they're working side by side, rather than, "Let me train you and here's a job and I'll just supervise and watch". (Chris, manager, November 2021)*

Working together generated a sense of empowerment and mutual respect among supported and non-supported employees.

At the WISE settings, supported employees were also matched with a 'Buddy' from the Support Team on their first day. The buddy provided support for orientation and

settling in, for example, protocols for arriving at work and leaving work. The buddy is a key figure in resilience sharing and may occupy a variety of roles (i.e., trainer, manager, co-worker, volunteer) in addition to being a workplace friend. This boundary-spanning figure shifts the power dynamic of support by embodying a workplace truth—that we all depend on colleagues to some degree for success in our respective jobs. However, further work is required in this area to develop appropriate funding approaches, including the use of NDIS funds.

Having the right team composition is key to individual experiences of success in open employment. As Mike explains, team members with different strengths provide learning opportunities and a sense of safety and are more efficient.

> *. . .it does make it more enjoyable, and more of a happier workplace. But it also makes it an efficient workplace as well. . . So it actually—you know, being a team player it actually helps it be comfortable as a person, but it also makes it more efficient. You get the job done better, and you also get the job done safer. And the more safe you are, the smarter you work.*

> *. . .there is even like there's a couple of jobs that I can do that another job I can't do as well, and then I can do that job for her and take over, and do that job for her. And then there's another girl that's—that I can't do the job as well, and she takes over from me. And we all help each other wherever we can.*

> *We've all got different strengths, and different needs, and what we can do and what we can't do. (Mike, supported employee, October 2021)*

Underpinning the acknowledgement of 'what we can do and what we can't do' is the team setting that allows for shared responsibility, understanding, acceptance, and a strengths-based workplace dynamic.

In addition to these approaches, something more is required: a hybrid approach to the field of employment that people with a disability operate in and shape. WISE support worker Jean (November 2021) was asked to describe what supports had been in place when someone living with a disability thrived in open employment. Jean reflected on the movement of individuals back and forth between open employment and the WISE. She noted that some people are in an in-between state: too confident for supported employment but not confident enough for long-term open employment, especially if that means giving up the relative security of disability supports from the government. In this context, the option for movement along different trajectories at different points in time is key.

## 4. Discussion

*Tailored Returns and Hybrid Employment for People Living with a Disability*

Resilience has been framed within neoliberal discourse as an individual trait and one that should be nurtured within the identity of individuals [10]. However, as King et al. [9] (pp. 4–5) remind us, 'responsibility and resilience are not exclusively neoliberal tropes. . . they have been put to use by actors from across the political spectrum', and 'both have a much longer history and wider reach than contemporary neoliberal discourses and policies'. In this paper, we have argued that resilience can be thought of as a dynamic or a shared trait between an organisation and an individual. This re-purposing of the word resilience takes place in a space of tension, at the intersection of advanced capitalist neoliberal logics (i.e., individualisation and responsibilisation) and supported employment 'network logics'.

Our research addressed the following questions: (a) what are the barriers people with a disability encounter in transitioning from supported to open employment, and (b) how does a large WISE operating in the disability sector support individual pathways into open employment? Our findings illustrate a range of barriers, including employer knowledge and confidence, lack of support and preparation/training, funding disincentives, and encounters of discrimination in open employment. Our findings were limited by the single case study design of the research project, which meant that data were collected at only one WISE.

Social enterprises, employers, and service providers share in the burden of responsibility carried by individuals to carve out an employment pathway in an increasingly complex and uncertain labour market shaped by ableism, stigma, funding barriers, and time-assessed productivity measures. Building networks and sustained support into the values that drive employment practices is a way of challenging norms that marginalise and exclude. WISEs can create networks and support structures with employers and service providers that generate the necessary organisational conditions for different forms of individual resilience to emerge.

WISE staff, supported employees, and employers in the open labour market are working together to address these barriers and challenge the 'ideal' and narrow conceptions of the worker. They do this by deploying a variety of mechanisms; for example, the allocation of buddies is used to span the boundaries of friend/co-worker and challenge hierarchical workplace cultures, particularly where managers take on the role of being a buddy. Additionally, customised and tailored employment identifies strengths in a range of contexts, and work conditions matched to individuals. The case study WISE bridged the gap to the open labour market through the development of working relationships within the community.

One of the 'neoliberal challenges' for WISEs more broadly is that there is an almost unconscious assumption inherent in the disability employment sector that the transition from supported to open employment is, ideally, a one-way street and, along the way, supports are gradually reduced to generate an 'ideal', independent worker [28]. It may be that the metaphor of a pathway closes possibilities and lateral movement for people with a disability. This has consequences for individual experiences of health and well-being, socio-economic independence, and financial sustainability. Rather than a one-way street, which is paved and straight, we suggest conceptualising pathways as multiple and multidirectional—a network of trails that may meander and double back as they adapt to the landscape. For people with a disability, the personal and employment landscape can be uneven and unpredictable. What is required is an employment approach that enables people to navigate this landscape. This is not a short, linear journey: it takes place over time and may look quite different at different points on the journey. This type of metaphor for disability employment pathways could allow us to help individuals create their own trails that alter the employment landscape, bringing into view new possibilities and choices.

Recent policy shifts in Australia aim to enhance the open employment opportunities of people with a disability and challenge the categories of 'open' and 'supported employment' by 'replacing these with a standard of high quality, inclusive employment that applies to all settings' [36] (p. 2). The challenge is the apparent lack of flexibility in the employment pathway for individuals who seek to move between WISE environments and open employment. More research is needed to understand how this movement can be best supported and the policy interventions that are required.

## 5. Conclusions

We are inspired by an early assessment of the purpose of WISEs offered by Davister, Defourny, and Gregoire [12] (p. 3) that social enterprises must incorporate a type of hybrid pathway or 'tailored follow-up' for their clients. The WISE involved in this study was able to check in on clients in the short term and provide follow-ups over the long term, but disability funding structures made this difficult. We suggest there is also a need for tailored returns that allow people living with a disability to sample and attempt open employment with the knowledge that it is socially and financially possible to return to more supported forms of employment. This movement shapes a hybrid employment landscape where both WISE and open employment opportunities become possible.

Tailored returns would embed in work plans the possibility that people living with a disability may wish to return to or continue part-time at a WISE to build new skills and confidence with the knowledge of what open employment entails. Barriers to tailored returns—such as funding structures that reduce disability supports for attempting open

employment—should be considered incongruent with the societal imperative to support people who live with a disability [36].

Further research is required to understand how employers in the open labour market can be supported with resources and knowledge to change the way meaningful and purposeful employment is made available to people with a disability. A shared resilience approach supports the development of employment trails that are co-created with employers, networks, and individuals. WISEs have a leading role to play in modelling these employment practices and shaping a new employment landscape.

**Author Contributions:** Conceptualization, P.C. and L.J.H.; methodology, P.C. and R.E.; formal analysis, P.C. and R.E.; investigation, P.C., E.W. and J.C.; resources, P.C.; data curation, P.C.; writing—original draft preparation, P.C. and L.J.H.; writing—review and editing, E.W., J.C., A.J. and R.E.; visualization, P.C.; supervision, E.W. and R.E.; project administration, P.C. and E.W.; funding acquisition, P.C., J.C. and E.W. All authors have read and agreed to the published version of the manuscript.

**Funding:** This research was funded by the Department of Social Services (DSS), Information Linkages and Capacity Building initiative grant number 4-G0PL3ZK.

**Institutional Review Board Statement:** This study was conducted in accordance with the Australian National Statement on Ethical Conduct in Human Research (2018) and approved by the Swinburne University of Technology Human Research Ethics Committee (protocol code 20215873-8392, 6 September 2021).

**Informed Consent Statement:** Informed consent was obtained from all subjects involved in the study.

**Data Availability Statement:** The data presented in this study are available on request from the corresponding author. The data are not publicly available due to privacy concerns.

**Acknowledgments:** We would like to acknowledge and thank our interview participants from diverse sectors and organisations who have generously contributed their time. We would like to thank the DSS Information Linkages and Capacity Building initiative and our 2021 partner organisation, which contributed significantly to the research project.

**Conflicts of Interest:** The authors declare no conflicts of interest. The funders had no role in the design of the study; in the collection, analyses, or interpretation of data; in the writing of the manuscript; or in the decision to publish the results.

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
