# Peer review of "The Role of Shared Resilience in Building Employment Pathways with People with a Disability"

_disabilities, doi:10.3390/disabilities4010008_

Round 1
Reviewer 1 Report
Comments and Suggestions for Authors
This is a clear and well-written paper. From the analysis of the entire work we can notice that the authors are well familiarized with the topic and literature. I appreciate that the authors highlight the fact that it is not enough for disabled to be supported to find a job, but it is far more important that they are helped to keep their job and remain employed. The results of this research are obviously useful for policy makers.
The article is a bit long and I think the authors can be more concise in presenting their results and recommendations. Is it necessary to present the impressions of people with disabilities who participated in ADE and who received support in the body of the article? I am not familiar with such presentations in the body of a scientific paper, but of course it also depends on the vision of the authors.
Overall, I find the article worth to be to be published. The obtained results and solutions are important to Disabilities Journal readers.
Author Response
We appreciate the reviewers detailed feedback and have addressed each concern in our paper.
We have edited and trimmed the manuscript to reduce the word count. We appreciate the perspective of the reviewer, and believe it is necessary to include the voices of people with a disability to share their experience of participation in the ADE and their employment journey.
Reviewer 2 Report
Comments and Suggestions for Authors
The paper is interesting and valuable. The authors analyze how people with disability can be supported into employment by an approach specific to Australian circumstances. A qualitative approach was used based on a case study concerning Work Integration Social Enterprise (WISE) model. The empirical research material comes from 27 interviews, 4 workshops and 5 steering committee meetings. Crucial issues from interviews were identified by a selective and iterative coding approach.
The strength of the study is that it was carried out with a participating organization which deals with supported employment. Action learning was involved and the research was collaborative in nature allowing for better understanding of the problems.
The discussed topic is of great importance for the societies and social inclusion. Exploring and assessing solutions for employing people with disabilities is an important scientific topic worth conducting in-depth research.
However in my opinion the paper may be improved in the following respects:
1. No research questions nor hypotheses are given directly in the paper. I think that presenting them at the beginning of the paper would facilitate the reader to follow the authors’ research concept and argumentation. In the introduction there are sentences which describe what is the objective of the study and what is explored and suggested but the aims of the study are not explicitly given. Is it exploration? Is it verification of the WISE model?
2. In the conclusion section no limitations of the study are presented. In my opinion they should be discussed.
3. According to journal guidelines: “References: References must be numbered in order of appearance in the text (including table captions and figure legends) and listed individually at the end of the manuscript.”. I seems that your paper does not follow this rule.
Author Response
Thank you, reviewer 2.
The research questions have been added at the start of the paper.
The limitation and next steps have been identified in the conclusion and discussion section.
References have been formatted to comply with journal requirements.
Reviewer 3 Report
Comments and Suggestions for Authors
I want to thank the authors for addressing an important topic. Please see specific suggestions for improvement below.
1. A credible definition of WISE should be introduced early in the manuscript with appropriate supporting references. I would also add that the authors should strive to rely less on unpublished self-citations when developing their key arguments – I would like to see more peer-reviewed sources from quality journals.
2. I would also encourage the authors to soften some of their claims. The authors should narrow the focus of the paper. The case study is not about PWD in general, it is about a proposed alternative employment pathway for people with high support needs who would ordinarily be likely to seek employment via an ADE.
3. While I am sympathetic to the need for new employment support models and agree that shared resilience is an important consideration, I am not sure that the evidence provided within this manuscript is strong enough to justify the claims made.
4. Notwithstanding the use of subjective and emotional language at times, which I encourage the authors to replace with more objective and unemotive phrasing, the initial conceptual development is good. I particularly like the discussion of an “ideal worker,” however, I would suggest that the authors focus only on disability as the information on other forms of inequality is distracting (and unnecessary).
5. The use of an action research methodology is well justified and appropriate. However, the presentation of the analysis was confusing and inconsistent with the coding approach described. The authors need to focus on synthesizing the data to identify key insights related to the themes, rather than providing an autobiographical summary of the experiences of different persons within the study (e.g., discussion of Max). The quotes should also be used more judiciously, focusing on strong illustrations of the key points to be made. At the moment, the analysis reads like a newspaper article rather than an academic paper.
6. The conclusion seems largely disconnected from the analysis. The authors also need to provide acknowledgement of limitations and a clearer articulation of the implications for theory and practice is needed.
Comments on the Quality of English Language
There were also a number of minor issues requiring attention:
a. I think that the embedding of quotes within the sub-headings is confusing and detracts from the clarity of the manuscript. Sub-headings are important signposts, so I would recommend removing the text before the colon.
b. I also would prefer that a section did not commence with a quotation (e.g., Patrice quote, p.8). It is best to situate the reader with an opening statement and then introduce a quotation in support of any claims made.
c. Lots of missing and incorrect references (e.g., Barraket et al., 2016), even a self-cite is incorrect (e.g., “Wilsom” on page 2). Please carefully check and correct the references.
d. The manuscript could also benefit from a good copyedit. While the language in general is fine, I did notice some oversights (e.g., missing words, clunky phrasing, orphan paragraphs) that could be attended to.
Author Response
Thank you, reviewer, 3.
A definition of WISE has been added with supporting references, most self-citations have been replaced with additional peer-reviewed sources.
Language and claims have been softened and the focus and clarity of the paper improved. The argument for resilience has been strengthened and developed to show its relevance to the employment experience of PWD.
The data section has been heavily edited. The key themes from the coding are woven into the data section and illustrated by the stories of PWD to acknowledge personal experiences and voice of PWD. The limitations and a clearer articulation of the implications for theory and practice have been identified in the discussion and conclusion section.
Round 2
Reviewer 3 Report
Comments and Suggestions for Authors
The authors should be commended for the revisions. There are still some minor typos (e.g., NIDS). A careful copyedit is recommended.
Comments on the Quality of English Language
See above.